# Vitamin D status in chronic fatigue syndrome/myalgic encephalomyelitis: a cohort study from the North-West of England

Kate E Earl,[1] Giorgos K Sakellariou,[1,2] Melanie Sinclair,[1] Manuel Fenech,[1,3] Fiona Croden,[4] Daniel J Owens,[5] Jonathan Tang,[6] Alastair Miller,[3] Clare Lawton,[4] Louise Dye,[4] Graeme L Close,[5] William D Fraser,[6] Anne McArdle,[1] Michael B J Beadsworth[1,3]

For numbered affiliations see end of article.

**Correspondence to**
Dr Michael B J Beadsworth;
mike.beadsworth@rlbuht.nhs.uk

## ABSTRACT

**Objective** Severe vitamin D deficiency is a recognised cause of skeletal muscle fatigue and myopathy. The aim of this study was to examine whether chronic fatigue syndrome/myalgic encephalomyelitis (CFS/ME) is associated with altered circulating vitamin D metabolites.

**Design** Cohort study.

**Setting** UK university hospital, recruiting from April 2014 to April 2015.

**Participants** Ninety-two patients with CFS/ME and 94 age-matched healthy controls (HCs).

**Main outcome measures** The presence of a significant association between CFS/ME, fatigue and vitamin D measures.

**Results** No evidence of a deficiency in serum total 25(OH) vitamin D ($25(OH)D_2$ and $25(OH)D_3$ metabolites) was evident in individuals with CFS/ME. Liquid chromatography tandem mass spectrometry (LC–MS/MS) analysis revealed that total 25(OH)D was significantly higher (p=0.001) in serum of patients with CFS/ME compared with HCs (60.2 and 47.3 nmol/L, respectively). Analysis of food/supplement diaries with WinDiets revealed that the higher total 25(OH) vitamin D concentrations observed in the CFS/ME group were associated with increased vitamin D intake through use of supplements compared with the control group. Analysis of Chalder Fatigue Questionnaire data revealed no association between perceived fatigue and vitamin D levels.

**Conclusions** Low serum concentrations of total 25(OH)D do not appear to be a contributing factor to the level of fatigue of CFS/ME.

## INTRODUCTION

Chronic fatigue syndrome (CFS), also referred to as myalgic encephalomyelitis (ME), is a debilitating and complex disease characterised by prolonged and disabling fatigue.[1] Patients with CFS/ME suffer from a range of symptoms including headache, painful lymph nodes, sore throat, difficulty in sleeping or insomnia, poor cognitive function, muscle/joint pain and postexertional

## Strengths and limitations of this study

► This is the first cross-sectional analysis of vitamin D status to be determined in patients with chronic fatigue syndrome/myalgic encephalomyelitis (CFS/ME).

► Patients were referred through a single National Health Service (NHS) CFS services unit at the Royal Liverpool and Broadgreen University NHS trust and diagnosed with CFS/ME using the National Institute for Health and Care Excellence guidelines ensuring diagnostic uniformity.

► Serum 25-hydroxyvitamin D, the main marker of vitamin D status, was determined using liquid chromatography tandem mass spectrometry and comprehensive dietary/supplement analyses were undertaken to support vitamin D serum data.

► Recruitment included a broad spectrum of ambulatory patients with CFS/ME, age and sex matched to health controls, but excluded the severest, non-ambulatory patients with CFS/ME.

malaise. CFS/ME can affect individuals of all ages, races and socioeconomic status. Prevalence of the disease has been estimated between 0.2% and 1%.[2–4] Despite the severity of CFS/ME and the prevalence of the disease, the aetiology and pathophysiology of CFS/ME is yet to be determined due to a lack of robust evidence within current literature to confirm any specific biological abnormalities. Current areas of investigation include infection, endocrine dysfunction, autonomic nervous system imbalance and altered immunity.

Vitamin D regulation has become a leading focus of health professionals following its implication in health and disease.[5–7] Vitamin D metabolites are regarded as vital endocrine regulators of bone health via their role in calcium and phosphate homeostasis.[8 9] Evidence has shown that vitamin D deficiency

$(25(OH)D<30\,nmol/L)$[10] is associated with numerous adverse health outcomes, including risk of rickets in children or osteomalacia in adults, increased risk of fractures, falls, infectious disease, type 1 and type 2 diabetes, cancer, autoimmune disease, multiple sclerosis, hypertension and heart disease.[11] Since the identification of the vitamin D receptor in almost all tissues of the human body including skeletal muscle,[12] evidence suggests that vitamin D metabolites are also implicated in skeletal muscle function and repair.[13–16] Clinical trials undertaken in geriatric and clinical populations[13 17 18] have established a potential link between vitamin D sufficiency, as assessed by circulating total 25-hydroxyvitamin D (25(OH)D) and physical performance including measures such as balance, gait speed, hand grip and lower limb strength, highlighting the role of vitamin D status in muscle function. In support of these findings, longitudinal studies undertaken in vitamin D-deficient adults showed that correcting vitamin D deficiency with use of dietary supplement interventions resulted in improved muscle mitochondrial function, associated with an improvement in symptoms of myopathy and fatigue.[19] The potential importance of vitamin D signalling is also highlighted by the ability of vitamin D signalling to modulate the activation of nuclear factor kappa B, a pleiotropic transcription factor involved in many biological and pathological processes including inflammation and immunity.[20] Indeed, $1,25(OH)_2D$ is also proposed to be a potent regulator of innate and acquired immunity.[21]

Due to the pleiotropic functions of vitamin D metabolites, the sterols have been examined in the context of disease and have consequently been correlated with a number of pathologies including tuberculosis, motor neuron and autoimmune diseases[22–24] and in clinical populations including patients with osteomalacia, who display skeletal muscle myopathy and weakness that is responsive to vitamin D supplementation.[25] Despite non-scientific media interest, few data exist examining circulating concentration of vitamin D metabolites in patients with CFS/ME. There has been only one retrospective study that suggests that patients with CFS/ME may experience suboptimal concentrations of circulating 25(OH)D,[26] hence the assumption that low circulating 25(OH)D concentrations in CFS/ME may be responsible for the altered immune profiles[27 28] and fatigue. The objective of the current study was to determine whether vitamin D status, as assessed by circulating 25(OH)D, is a contributing factor to the pathophysiology of CFS/ME. To meet this objective, we investigated whether CFS/ME was associated with altered serum concentrations of vitamin D metabolites.

## METHODS

### Recruitment

Male and female patients with CFS/ME (n=92) were recruited through the regional National Health Service CFS services based in Liverpool at the Royal Liverpool and Broadgreen University Hospital Trust, Merseyside, UK. All patients were newly diagnosed by clinicians in accordance with the recommended National Institute for Health and Care Excellence (NICE) guidelines 2007[29] and covered a wide spectrum of disease severity from mild, through to moderate and severe CFS/ME. All patients routinely completed the Chalder Fatigue Questionnaire. Bed-bound/non-ambulatory patients were not enrolled in the study. Ninety-four age-matched and sex-matched healthy controls (HCs) (with exclusions of smoking or substance misuse, medication including anti-inflammatory medication or non-ambulatory subjects) were also recruited through local advertisement in the city of Liverpool and during CFS/ME recruitment sessions and served as the control group. This group also completed the Chalder Fatigue Questionnaire. Sample size was determined according to power calculations to allow for a significant difference in serum 25(OH)D to be detected between the two cohorts. Individuals were excluded from the study if they smoked, had a history of substance misuse or had any underlying inflammatory-based medical conditions or treatments. This study conformed to the standards set by the Declaration of Helsinki, and the procedures were approved by the local University of Liverpool peer review committee and multicentre research ethics committee. All participants gave informed written consent for their participation.

### Sample collection

For the analysis of vitamin D metabolites including $25(OH)D_2$ and $25(OH)D_3$, participants provided a venous blood sample collected from the antecubital vein into two serum separator (4.7 mL) vacutainers. Blood samples were incubated for 45 min at room temperature to induce clotting, followed by centrifugation at 1500 g for 15 min at 4°C. Serum was aspirated from the samples and stored at −80°C until analysis. All serum samples were given an anonymous coded identification number, which was sequentially generated by the study team.

### Analysis of vitamin D

Vitamin D metabolites $25(OH)D_2$ and $25(OH)D_3$ were analysed as previously described.[8] $25(OH)D_2$ and $25(OH)D_3$ were extracted from serum samples, following zinc sulfate protein precipitation, using Isolute C18 solid phase extraction cartridges. Potential interfering compounds were removed by initial elution with 50% methanol followed by elution of the vitamins using 10% tetrahydrofuran in acetonitrile. Dried extracts were reconstituted prior to injection into a high-performance liquid chromatography tandem mass spectrometry (LC–MS/MS) in the multiple reaction mode. The multiple reaction mode transitions (m/z) used were 413.2>395.3, 401.1>383.3 and 407.5>107.2 for $25(OH)D_2$ and $25(OH)D_3$, and hexadeuterated $(OH)D_3$, respectively. The assay was validated against published acceptance criteria (FDA 2001). Assay sensitivity was determined by the lower limit of quantification: $25(OH)D_3=2.5\,nmol/L$ and 25(OH)

$D_2=2.5$ nmol/L. Coefficients of variation for the assay were 10% across a working range of 2.5–625 nmol/L for both $25(OH)D_2$ and $25(OH)D_3$.

Intra-assay precision was assessed by running quality control materials (Chromsystems Instruments and Chemicals, Gräfelfing, Germany) and commercially bought material (calf serum) 10 times within a single run, and separately over 15 runs for interassay assessment. Extraction recovery was assessed by determining the amount of vitamin $D_3/D_2$ recovered from the amount spiked prior to extraction. The percentage recovery was calculated by the measured value against the sum of endogenous value and spiking concentration. Spiked recovery is determined by adding known quantity of $25(OH)D_3/D_2$ to serum samples with different concentrations of endogenous $25(OH)D$. MassCheck calibration materials and controls are traceable against NIST 972 reference material.

The LC–MS/MS method used in the present study has previously been validated against other commercially available assays and is regarded as the most valid and reliable technique for the assessment of vitamin D metabolites including $25(OH)D_2$ and $25(OH)D_3$.[30]

### Dietary/supplement analyses

Structured open-ended food/supplement diaries were provided to all patients and control. All participants were informed on how to complete the food record. The diary also contained a written example for reference. During a 7-day period, all consumed foods/drinks and supplements had to be reported with notification of time of consumption, estimated consumed quantity expressed as a household measure, unit or weight, specification and if present, a brand name. Food diaries were analysed using WinDiets Research Version (2010) software, written by Dr Alan Wise, The Robert Gordon University, Aberdeen.

### Patient involvement

The development of both the study research question and outcome measures took into careful consideration the needs and impact on the patient cohort particularly given the highly sensitive nature of this group. Careful consideration was taken over the execution of study visits and the impact of the study protocol and sample collection. The study was designed to reduce the impact and disturbance to the patient's condition. Senior specialists in CFS/ME provided advice with regard to study design. No patients were involved in setting the research question, nor were they involved in developing plans for design or implementation of the study. No patients were asked to advise on interpretation or writing up of results. Registered patient support groups within the local area of Merseyside including patients with CFS/ME and their carers provided support in the recruitment of patients into the study through the dissemination of study information and recruitment. Results will be disseminated by presentation to local and national research and patient groups.

**Table 1** Baseline characteristics of study participants

|  | HC | CFS/ME |
|---|---|---|
| Women (n) | 58 | 66 |
| Men (n) | 36 | 26 |
| Caucasian (n) | 89 | 91 |
| Age (years) | 33±10 | 40±10 |
| BMI (kg/m$^2$) | 24.4±3.4 | 26.0±4.2 |
| Illness duration (months) | N/A | 36±4.8 |
| Sample acquisition | 94 | 92 |

Data are presented as mean±SD unless otherwise stated.
BMI, body mass index; CFS/ME, chronic fatigue syndrome/myalgic encephalomyelitis; HC, healthy control.

### Statistics

Data are presented as mean±SD throughout. The sample distribution was determined using the Pearson's coefficient indicating a predominantly parametric distribution. Single comparisons between two groups of normally distributed data were undertaken using independent Student's t-tests. For multiple comparison analyses to examine the effect of season, one-way analysis of variance followed by post hoc Tukey's LSD test was used. A p value <0.05 was considered to be statistically significant. SPSS V.22 was used to analyse all data. Figures were constructed with GraphPad Prism V.6.

### RESULTS

Details of the study population are presented in table 1. Epidemiological studies have reported that the prevalence of CFS/ME is significantly higher in women than in men[31] and the majority of patients with CFS/ME who took part in the present study were female (66 women and 26 men). Body mass index analyses revealed no significant differences between patients with CFS/ME and HCs (26.0 and 24.4 kg/m$^2$, respectively) (table 1). Average duration of illness in the patients with CFS/ME was 36±4.8 months.

Total serum $25(OH)D$ was assessed by measuring the sum of $25(OH)D_2$ and $25(OH)D_3$ metabolites. Circulating $25(OH)D_2$ was undetectable in 86% and 89% of CFS/ME and HC individuals, respectively (data not shown). Undetectable concentrations of circulating $25(OH)D_2$ in human samples have previously been reported in a number of studies.[32] Total serum $25(OH)D$ concentration was significantly higher (p=0.001) in patients with CFS/ME compared with HCs (60.2 and 47.3 nmol/L, respectively) (figure 1A).

To further examine the observed difference in vitamin D status between patients with CFS/ME and HC, the effects of seasonal variation in circulating concentrations of $25(OH)D$ were determined. Statistical analyses revealed a significant interaction between group and season for total serum $25(OH)D$ (figure 1B). Patients with CFS/ME demonstrated significantly higher (p=0.01) total serum $25(OH)D$ concentrations during the autumn and

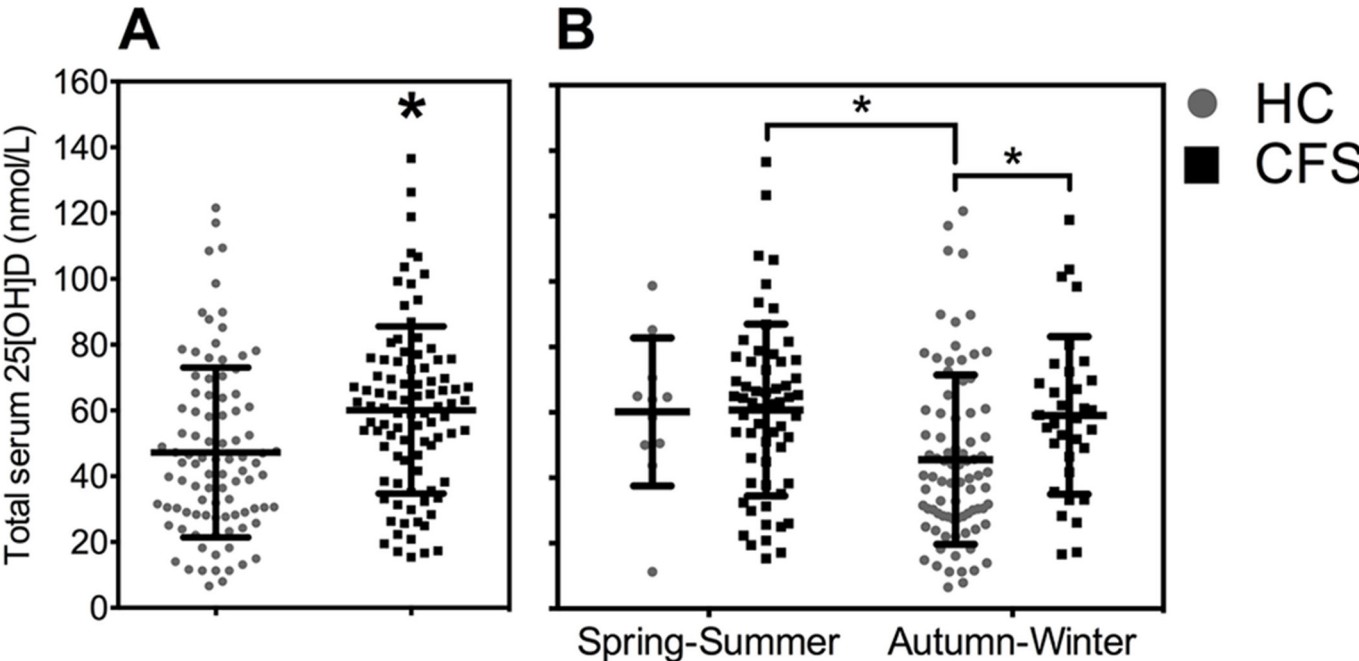

**Figure 1** (A) Distribution plots of individual circulating total 25(OH)D concentrations (nmol/L) assessed in patients with chronic fatigue syndrome/myalgic encephalomyelitis (CFS/ME) and healthy controls (HCs) using liquid chromatography tandem mass spectrometry. *p<0.05 compared with values from the HC. (B) Total serum 25(OH)D concentration in patients with CFS/ME and HCs recruited during the spring–summer months (1 March31 August) and autumn–winter months (1 September–28 February). *p<0.05 compared with values from the HC group recruited during autumn/winter. #p<0.05 compared with values from the HC group recruited during the Spring/Summer months. Close grey circles denote HC and closed black boxes denote CFS/ME. Each data point refers to an individual participant. Solid black bars represent the mean±SD.

winter months (autumn/winter) compared with HC individuals (59.1±24.1 and 45.6±25.9 nmol/L, respectively), with no significant difference evident during the spring and summer months (spring/summer) (figure 1B). The average total serum 25(OH)D concentration from HCs recruited during spring/summer tended to be ~30% higher (p=0.08). Although this did not reach statistical significance in comparison to individuals recruited in autumn/winter (60.2±22.7 and 45.6±25.9 nmol/L, respectively), (figure 1B), these data are in agreement with previous clinical studies demonstrating that vitamin D status varies seasonally, with concentrations of 25(OH) D increasing in spring and peaking at summer.[33]

The higher total serum 25(OH)D concentrations in patients with CFS/ME during the autumn/winter months is likely to be a result of increased vitamin D intake. To address this possibility, dietary/supplement assessment of vitamin D intake from both diet and in the form of supplements was undertaken although, due to compliance issues, these data were only provided by 21 HCs and 24 patients with CFS/ME (table 2). Dietary intake of vitamin D did not differ significantly between patients with CFS/ME and HC individuals. However, more patients with CFS/ME reported multivitamin supplement and/or vitamin D supplement consumption than HCs. Because of the increased use of supplements by patients with CFS/ME, the cohort of patients with CFS/ME showed higher (p=0.05) average vitamin D intake through use of supplements compared with the cohort of HCs (4.9±9.7 and

0.6±1.7 µg/day, respectively) (table 2). When we limit the analysis to those individuals taking vitamin D supplements, the average intake for HCs was 4.2 µg/day compared with 14.1 µg/day for patients with CFS/ME. In addition, analysis revealed no differences in total serum 25(OH)D in the patients with CFS/ME compared with HC individuals reporting no vitamin D supplement intake (figure 2). Overall these findings suggest that the higher serum 25(OH)D concentrations observed in patients with CFS/ME during the autumn and winter months was likely to be due to increased vitamin D intake through use of supplements. Individuals with CFS/ME reported significantly higher levels of fatigue than the HC group in the Chalder Fatigue Questionnaire; a widely used and validated health questionnaire (data not shown in detail). To assess whether serum concentrations of total 25(OH)

**Table 2** Dietary/supplement assessment of average vitamin D intake by patients with chronic fatigue syndrome/ myalgic encephalomyelitis (CFS/ME) compared with healthy control (HC) individuals

|  | HC | CFS/ME | p Value |
|---|---|---|---|
| Average vitamin D intake—diet (µg/day) | 2.9±2.1 | 2.4±1.6 | 0.6 |
| Average vitamin D intake—supplements (µg/day) | 0.6±1.7 | 4.9±9.7 | 0.05 |

Data are presented as mean ±SD.

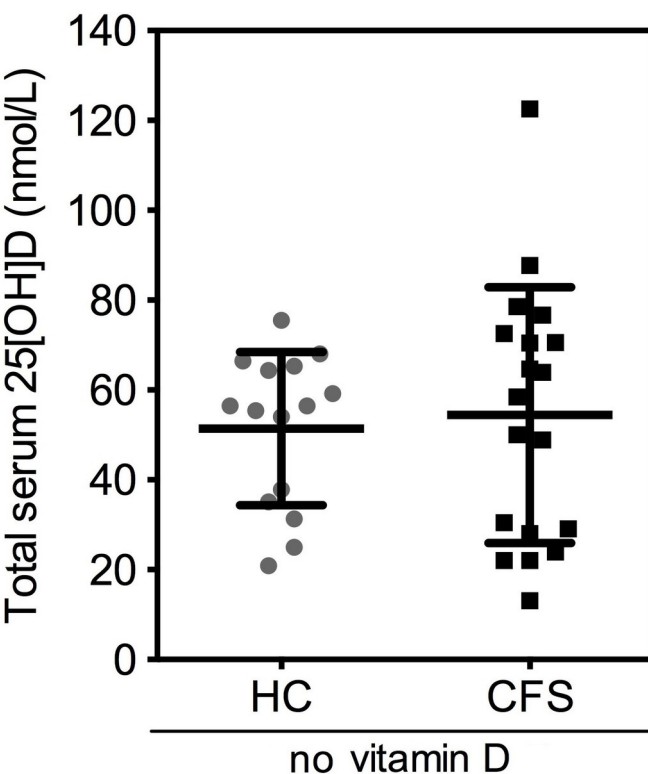

**Figure 2** Total serum 25(OH)D concentration in patients with chronic fatigue syndrome/myalgic encephalomyelitis (CFS/ME) and healthy controls (HCs) who reported no intake of vitamin D supplements. Each data point refers to an individual participant. Solid black bars represent the mean±SD.

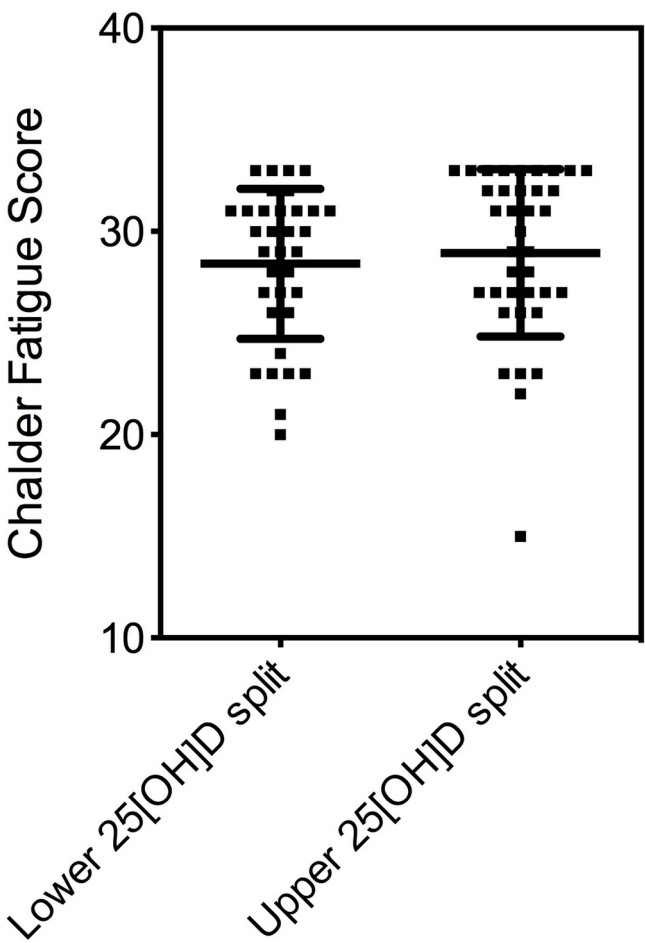

**Figure 3** Chalder Fatigue Scores in patients with chronic fatigue syndrome/myalgic encephalomyelitis (CFS/ME). 25(OH)D values from the CFS/ME group were ordered from lowest to highest and split equally into an upper and lower range. Each data point refers to an individual participant. Solid black bars represent the mean±SD.

D were a contributing factor to the level of fatigue in CFS/ME, 25(OH)D values from the CFS/ME group were ordered from lowest to highest and split equally into an upper and lower range (figure 3). The data presented show no evidence of a difference in mean or distribution of Chalder Fatigue Questionnaire Score between the two groups and so suggest that total serum 25(OH)D concentration does not appear to be a major contributing factor to the level of fatigue in patients with CFS/ME.

## DISCUSSION

The present findings suggest that patients with CFS/ME do not exhibit insufficient concentrations of circulating total 25(OH)D, but the values are significantly higher compared with HCs. The average concentration of serum total 25(OH)D in patients with CFS/ME was 60.2 nmol/L and was greater than the sufficient cut-off of ≥50 nmol/L set by the UK and US Institutes of Health.[33] Scores from the Chalder Questionnaire revealed no differences between individuals with CFS/ME with high or low total 25(OH)D concentrations. This finding implies that vitamin D deficiency is not a major contributing factor to the pathophysiology of CFS/ME. The data from the present study further demonstrated that the average concentration of circulating total 25(OH)D in the HC group

was lower (47.3 nmol/L) than the suggested sufficient concentrations (>50 nmol/L), in agreement with recent reports from the NICE that insufficient serum vitamin D concentrations are common in the UK. The present findings suggest individuals with CFS/ME have higher total 25(OH)D than HCs, particularly in the spring/summer months, due to the consumption of vitamin D supplements. In contrast, further analysis revealed no significant differences in total 25(OH)D between individuals with CFS/ME and HCs who reported no supplement intake. Overall these findings imply that the higher total 25(OH)D in the CFS/ME group is likely attributed to the overall higher supplemental vitamin D intake.

Individuals with circulating 25(OH)D concentrations >50 nmol/L are likely to have sufficient concentrations to maintain normal biological functions. However, concentrations between 30 and 50 nmol/L are considered to be insufficient and <25 nmol/L are likely to lead to the pathology of muscle and bone disorders such as osteomalacia and osteoporosis and to reduce overall health.[34] Previous research has been undertaken to determine

changes in total 25(OH)D throughout the year as the intensity and hours of sunlight change.[35] The current study further sought to examine the changes that occur in vitamin D status throughout the year in both patients with CFS/ME and age-matched/sex-matched HC individuals. Seasonal variation analyses revealed that HC individuals recruited during autumn/winter exhibited insufficient levels of circulating 25(OH)D (<50 nmol/L) with concentrations rising significantly to normal values (>50 nmol/L) in individuals recruited during spring/summer. In contrast, there was a lack of difference in circulating 25(OH)D between autumn/winter and spring/summer in patients with CFS/ME. Data showed an ~50% reduction in vitamin D supplement intake in the patient cohort recruited during the summer months. Caution is required when interpreting these findings since the study is cross-sectional and so represents different patients and these changes in supplement intake did not reach significance (data not shown), but may suggest that patients with CFS/ME reduce their supplement intake of vitamin D during the summer months. Alternatively, the patients may not have been exposed to increased sunlight during the summer months due to reduced mobility and being housebound as a result of the symptoms associated with CFS/ME. The observed suboptimal circulating 25(OH)D concentration in healthy individuals during the autumn and winter months was likely due to insufficient vitamin D intake and/or lack of adequate exposure to sunlight.[13] The higher prevalence of sufficient 25(OH)D concentrations observed in patients with CFS/ME during autumn/winter were associated with higher vitamin D intake through use of supplements when compared with HCs.

Patients with CFS/ME suffer from a wide range of debilitating symptoms and, depending on these symptoms, there may be reduction in exposure to the sun. The current study included patients across the disease spectrum from mild severity through to moderate and severe, although excluding bed-bound/non-ambulatory patients. As a result, it is plausible to assume that there may be a subgroup of patients with CFS/ME, who may be at greater risk for deficiency. One limitation of the present work was that the majority of participants who took part in the study, including CFS/ME and HC volunteers, were Caucasians who lived in the North-West of England hence the findings from the present study might not reflect all CFS/ME and HC populations. It is noteworthy to mention that there is evidence that vitamin D metabolism varies in different ethnic groups.[34] Future studies comparing CFS/ME including non-ambulatory patients and HC individuals from different origin, race and disease duration are also warranted. A further limitation to the study was that only the main marker of vitamin D status, that is, 25(OH)D, was measured. It has recently been shown that there is a need to assess all of the vitamin D metabolites.[35] It may be more appropriate to measure the amount of serum-free and bioavailable 25(OH)D as a predictor of vitamin D status than serum total 25(OH)D, while controlling for vitamin D binding protein phenotype.[36] Future studies assessing these aspects of vitamin D metabolism in patients with CFS/ME are warranted.

In conclusion, patients with CFS/ME do not exhibit insufficient concentrations of circulating $25(OH)D_3$ at any time of the year. Given that this cohort of patients with CFS/ME clearly displays symptoms, regardless of serum 25(OH)D concentration, these data indicate that vitamin D status may not be a major contributing factor to the pathophysiology of patient with CFS/ME symptoms, although further analyses of those individuals with lower circulating 25(OH)D and more detailed investigation into vitamin D metabolism is still warranted. Current health guidelines in the UK state supplementation of vitamin D should be recommended to individuals with increased risk of deficiency and the findings from the present study suggest, in agreement with other studies, that a significant percentage of healthy individuals exhibit insufficient concentrations of circulating vitamin D during the autumn and winter months. The current findings do not support the notion that vitamin D concentration contributes to the continuing symptoms of CFS/ME. However, it may be important for patients with CFS/ME to supplement with vitamin D during the winter months as it is well documented that non-supplemented individuals may typically present with insufficient serum 25(OH)D during the winter season.

**Author affiliations**

[1]Department of Musculoskeletal Biology, Institute of Ageing and Chronic Disease, MRC-Arthritis Research UK Centre for Integrated Research into Musculoskeletal Ageing, University of Liverpool, Liverpool, UK
[2]GeneFirst Ltd, Culham Science Centre, Oxfordshire, UK
[3]Tropical and Infectious Disease Unit, Royal Liverpool University Hospital, Liverpool, UK
[4]Human Appetite Research Unit, School of Psychology, University of Leeds, Leeds, UK
[5]Research Institute for Sport and Exercise Science, Liverpool John Moores University, Liverpool, UK
[6]Department of Medicine, Norwich Medical School, Faculty of Medicine and Health Sciences, University of East Anglia, Norwich Research Park, Norwich, UK

**Acknowledgements** We thank Colette Foster, Emmanuel Nsutebu and Libbe Ratcliffe, the Liverpool CFS/ME Therapy Service and the Royal Liverpool Clinical Research Unit for their support.

**Contributors** AM, MBJB, CL, LD and GLC conceived and planned the research. KEE, GKS, MS, MF, FC, DJO and JT carried out the research, supervised by AM and MBJB and funded by grants awarded to AM, CL, LD and MBJB. KEE and FC performed the analysis of the intake diaries. KEE, GKS and JT analysed and presented the data. KEE and GKS wrote the paper. All authors revised the paper.

**Funding** This study was supported by grants from the British Medical Research Council, the ME Association and The University of Liverpool.

**Competing interests** None declared.

**Patient consent** Obtained.

**Ethics approval** This study conformed to the standards set by the Declaration of Helsinki, and the procedures were approved by the local University of Liverpool peer review committee and multi-centre research ethics committee.

**Provenance and peer review** Not commissioned; externally peer reviewed.

**Data sharing statement** No additional data are available.

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
