## [Reviewer comments · BMJ Open]

ARTICLE DETAILS

TITLE (PROVISIONAL)	Vitamin D status in Chronic Fatigue Syndrome/Myalgic Encephalomyelitis: a cohort study from the North-West of England
AUTHORS	Earl, Kate; Sakellariou, Giorgos; Sinclair, Melanie; Fenech, Manuel; Croden, Fiona; Owens, Daniel; Tang, Jonathan; Miller, Alastair; Lawton, Clare; Dye, Louise; Close, Graeme; Fraser, William; McArdle, Anne; Beadsworth, Michael

VERSION 1 - REVIEW

REVIEWER	Esther Crawley University of Bristol, UK
REVIEW RETURNED	31-Jan-2017

GENERAL COMMENTS	This is a simple but important paper for patients with CFS/ME A few minor points 1. In the strengths and limitations section you may want to reconsider your description "mild to severe ambulatory subjects" as severe CFS/ME is generally regarded as being non ambulatory. I suspect you mean that they came to the clinic? Maybe just be clear on that.2. Same section - what do you mean by "under NICE guidelines". I think you mean, patients were diagnosed with CFS/ME following NICE guidance?2. In the introduction, I would give the range of the prevalence which is about 0.4 to the higher range (can you double check your figure of 3.29%) which is very high indeed. There is a systematic review on prevalence which gives about 0.6 from memory. I would use the systematic review or a range.3. Please remove funding by the MEA from "Patient involvement" to the funding section. Funding by a charity is not the same as patient input in to the study. This section is strange and looks like it might have been cut and pasted from something else. It is not usual to say you are going to report conferences within a paper. Can you please look at the Patient Involvement section.
---

REVIEWER	Jos WM van der Meer Dept of Medicine Radboud university medical centre Nijmegen The Netherlands
REVIEW RETURNED	14-Feb-2017

GENERAL COMMENTS	The authors have investigated the Vitamin D status in CFS. Although the rationale for the study is not very strong, the investigation was worth the effort. No difference was found between patients and controls, and that is a relevant answer. Given the negative result of the study, the paper could be shortened considerably in all its sections. Additional comments  1. The conclusions could be made a bit less prudent. I would not be inclined to further look into vitamin D (as indicated on page 11 line 13). Likewise a phrase like "these data indicate that vitamin D status may not be a major contributing factor" (page 11 line 18) is too prudent. One might say that there is no convincing reason for CFS patients to take Vitamin D supplements (as is apparently done quite commonly). 2. I would not use an artificial split for Vitamin D status in figure 3, but rather use the actual Vitamin D concentrations against the Chalder score. 3. Do not start the discussion with a new introduction. 4. Minor: change all subjects (page 6 line 27) into all patients and controls.
--

REVIEWER	Samantha Johnston Griffith University, Australia
REVIEW RETURNED	21-Feb-2017

GENERAL COMMENTS	Many thanks authors for your manuscript. The question is of clinical relevance to CFS/ME and indication of whether Vitamin D deficiency needs to be taken into consideration in this patient set. In the abstract and manuscript I would describe the clinical significance of this study in terms of objectives and conclusions. Placing it in a greater context of, if your study did or did not ID low vitamin D as an issue, and your recommendations in terms of considering this when examining patients with CFS/ME as I believe this is a question that would come up often. In its current form, the manuscript does require significant clarification and consistency. The initial aims, methodology, and outcome measures are not clearly defined. Eg. Information appears in results that isn't explained methodology. I highly recommend consulting STROBE guidelines for reporting case control studies, as recommended by BMJ Open as this will sharpen the impact of your study. Though you have completed the checklist, i do find the study has not addressed all areas. Please refer to my specific suggestions: Abstract Design:
---

	The study is a traditional case control study, not a traditional cohort study that would imply longitudinal measures Outcome measures: I would better define how vitamin D deficiency is measured. This would to me, form the primary outcome. Not the association itself (I would refer to that as the analysis) Results: I would rephrase the first sentence due to double negative. Methodology is present in the results which I would summarise separately ie. mass spec. further what dietary/supplement analyses was performed? Conclusions: Level of fatigue appears, however this was not defined or described as an outcome measure or as part of your aims. Strengths and limitations: It is now described as a cross-sectional analysis of CFS, however you should summarise imo that it was a case control study and carry this throughout the paper. You refer to how patients were referred and I would consider including this as part of your study design eg. a primary care sample was examined in your abstract. Introduction Reference 2 is incorrect, as this study only refers to case definitions used. The study you are referring to is Johnston et al. 2013 meta-analysis. The more appropriate statistic from this paper is 0.76% (95% CI: 0.23–1.29) for clinically assessed cases of CFS. The 3.29% was summarising self-reported cases and is thus, quite high. The objective is clearly written here though I would tend to interpret this the other way around ie. . The aim of the study would be see if vitamin D deficiency is associated with these patients. To achieve this you would complete the objective of examining circulating vitamin D... between cases and controls. Again the appropriate study design needs to be acknowledged. Methods Recruitment: I would also state how you defined a “healthy control”. Different studies have different inclusion criteria so I would make this clear in your study design. The Chalder fatigue questionnaire is not mentioned anywhere in the methodology. In recruitment, any use of questionnaires or other methods of data collection need to be outlined. Ethics number for project Commend the authors for clarify of methodology to examine LC-MS/MS. Patient involvement: This section appears misplaced in methodology and is more appropriate for the discussion. Issues of consent, and blinding are described in the current methodology seems sufficient. The considerations that were taken to include these patients as you have described imo that you describe here would be added to the discussion perhaps under a subheading such as “strengths and limitations of the study”... again refer to STROBE guidelines. Statistics: Not population distribution, sample distribution. These have different meanings due to the study design. I would again, refer to case vs. control comparison to make your study design clear. Results:
--	--

	Table 1 I would add a statistical comparison between groups The statement “However, more CFS/ME patients who provided 7-day dietary feedback reported multivitamin supplement and/or vitamin D supplement consumption than HCs and hence CFS/ME patients showed higher ($p = 0.05$) vitamin D intake through use of supplements compared with HC individuals (4.9 ± 9.7 and $0.6 \pm 1.7\mu\text{g/day}$, respectively), (Table 2).” Is not clear and needs rephrasing. I.e. more CFS patients completed the diary compared to controls... but then CFS patients had higher VIT D intake... shouldn't this be controlled.. i.e. you would exclude all those that didn't complete the food diary for that part of the analysis. These findings start to become a bit confusing and need clarification (1) what data you were able to collect. And then (2) how you analysed this. Attempting to summarise both in the same sentences etc. I think is what causes the confusion. Again this would benefit in methodology if you clearly list what all your outcome measures are (i.e. data you collected). Discussion: Avoid referral to “sufferers” in reporting as this is considered discriminatory writing in scientific reporting. More appropriate is patients or individuals. I think the main point is that even though this initial study has shown any differences between cases and controls, vitamin D deficiency should still be considered at the individual level when it comes to managing patient. I think this point really needs to be drawn out. I.e. once you really do clarify that aim and make it consistent. These points will ring clear.
--	---

VERSION 1 – AUTHOR RESPONSE

Reviewer 1:

A few minor points

1. In the strengths and limitations section you may want to reconsider your description "mild to severe ambulatory subjects" as severe CFS/ME is generally regarded as being non-ambulatory. I suspect you mean that they came to the clinic? Maybe just be

2. Same section - what do you mean by "under NICE guidelines". I think you mean, patients were diagnosed with CFS/ME following NICE guidance?

The two key points raised by the reviewer within the strengths and limitations section have been addressed and the manuscript has been altered accordingly (Page 2).

- Specifically, clarification regarding the severity of the cases recruited. The authors accept that the severest sufferers of CFS/ME are non-ambulatory. The study recruited a broad spectrum of ambulatory only patients with CFS/ME and we have removed the term 'mild to severe'.

- The reviewer correctly pointed out that patients were diagnosed with CFS/ME following NICE guidance. The text within the manuscript has been altered accordingly.

In the introduction, I would give the range of the prevalence which is about 0.4 to the higher range (can you double check your figure of 3.29%) which is very high indeed. There is a systematic review on prevalence which gives about 0.6 from memory. I would use the systematic review or a range. We agree with the reviewer that a range of prevalence would be a more suitable way of presenting CFS prevalence given the differing values across different publications. The manuscript has been changed accordingly (Page 3).

4. Please remove funding by the MEA from "Patient involvement" to the funding section. Funding by a charity is not the same as patient input in to the study. This section is strange and looks like it might have been cut and pasted from something else. It is not usual to say you are going to report conferences within a paper. Can you please look at the Patient Involvement section.

The paragraph regarding patient involvement was inserted in line with publication requirements by the BMJ but has been modified to take into consideration the reviewer's comments (Page 6).

Reviewer 2:

General comment: Given the negative result of the study, the paper could be shortened considerably in all its sections.

The authors have acknowledged this comment and have made some reductions, but feel the paper is at an appropriate length to include the necessary background and study information and cover the BMJ checklist. We agree that this is a negative finding but as such we also feel with current discussions regarding vitamin D, particularly in this patient group, it is an important manuscript.

1. The conclusions could be made a bit less prudent. I would not be inclined to further look into vitamin D (as indicated on page 11 line 13). Likewise, a phrase like "these data indicate that vitamin D status may not be a major contributing factor" (page 11 line 18) is too prudent. One might say that there is no convincing reason for CFS patients to take Vitamin D supplements (as is apparently done quite commonly).

The authors thank the reviewer for their comment, we do not feel it would be appropriate to suggest that CFS patients should not be taking Vitamin D supplements given the evidence that healthy control subjects reported inadequate circulating levels of vitamin D in the winter months. The authors feel it more appropriate to maintain their conclusions that vitamin D status is not a contributing factor in the symptoms associated with CFS/ME given the evidence presented.

2. I would not use an artificial split for Vitamin D status in figure 3, but rather use the actual Vitamin D concentrations against the Chalder score.

The authors felt it appropriate to split to data into two groups demonstrating the lower and upper values for Vitamin D (Page 8) but also plotted these data such that distribution could be visualised. Clearly there was no evidence for any difference in either the mean values for Chalder Fatigue or in the distribution and so further analyses was not seen to be warranted.

3. Do not start the discussion with a new introduction.

The authors acknowledge that this section of text may be more appropriately placed in the introductory section of the manuscript and have thus moved it to Page 3.

4. Minor: change all subjects (page 6 line 27) into all patients and controls.

In line with the reviewers comment above, the authors have altered all subjects to all patient and controls throughout the manuscript.

Reviewer 3:

The question is of clinical relevance to CFS/ME and indication of whether Vitamin D deficiency needs to be taken into consideration in this patient set. In the abstract and manuscript, I would describe the clinical significance of this study in terms of objectives and conclusions. Placing it in a greater context of, if your study did or did not ID low vitamin D as an issue, and your recommendations in terms of considering this when examining patients with CFS/ME as I believe this is a question that would come up often.

In its current form, the manuscript does require significant clarification and consistency. The initial aims, methodology, and outcome measures are not clearly defined. Eg. Information appears in results that isn't explained methodology. I highly recommend consulting STROBE guidelines for reporting case control studies, as recommended by BMJ Open as this will sharpen the impact of your study.

Though you have completed the checklist, I do find the study has not addressed all areas. Please refer to my specific suggestions:

We thank the reviewer for their comments however the authors do not agree with the major points made regarding the presentation of the manuscript. The authors remain content with the study design and presentation of this manuscript.

In brief, the study was designed and conducted as a cohort study and therefore we do not think the STROBE guidelines would be appropriate. The authors believe they have clearly defined all sections of the study and adhered to BMJ requirements during the writing of this manuscript, see below.

Abstract

Design:

The study is a traditional case control study, not a traditional cohort study that would imply longitudinal measures

Outcome measures:

I would better define how vitamin D deficiency is measured. This would be the primary outcome. Not the association itself (I would refer to that as the analysis)

This has been clarified.

Results:

I would rephrase the first sentence due to double negative. Methodology is present in the results which I would summarise separately ie. mass spec. further what dietary/supplement analyses was performed?

The double negative has been removed and the supplement analysis added.

Conclusions:

Level of fatigue appears, however this was not defined or described as an outcome measure or as part of your aims.

This has now been added.

Strengths and limitations:

It is now described as a cross-sectional analysis of CFS, however you should summarise in that it was a case control study and carry this throughout the paper.

You refer to how patients were referred and I would consider including this as part of your study design eg. a primary care sample was examined in your abstract.

We do not feel that this is necessary.

Introduction

Reference 2 is incorrect, as this study only refers to case definitions used. The study you are referring to is Johnston et al. 2013 meta-analysis. The more appropriate statistic from this paper is 0.76% (95% CI: 0.23–1.29) for clinically assessed cases of CFS. The 3.29% was summarising self-reported cases and is thus, quite high.

The authors thank the reviewer for pointing out this error and we have adjusted the manuscript in line with comments made on this sentence by Reviewer 1.

The objective is clearly written here though I would tend to interpret this the other way around ie.. The

aim of the study would be see if vitamin D deficiency is associated with these patients. To achieve this you would complete the objective of examining circulating vitamin D... between cases and controls. Again the appropriate study design needs to be acknowledged. We feel that the objectives are clear.

Methods

Recruitment:

I would also state how you defined a "healthy control". Different studies have different inclusion criteria so I would make this clear in your study design.

Exclusion criteria for HCs has been added (page 4).

The Chalder fatigue questionnaire is not mentioned anywhere in the methodology. In recruitment, any use of questionnaires or other methods of data collection need to be outlined.

This has now been added (Page 4).

Ethics number for project

We are not aware that this is needed.

Commend the authors for clarify of methodology to examine LC-MS/MS.

Patient involvement:

This section appears misplaced in methodology and is more appropriate for the discussion. Issues of consent, and blinding are described in the current methodology seems sufficient. The considerations that were taken to include these patients as you have described imo that you describe here would be added to the discussion perhaps under a subheading such as "strengths and limitations of the study"... again refer to STROBE guidelines.

This section has been modified in line with the comments of Reviewer 1.

Statistics:

Not population distribution, sample distribution. These have different meanings due to the study design.

I would again, refer to case vs. control comparison to make your study design clear.

Population distribution has been changed to sample distribution.

Results:

Table 1 I would add a statistical comparison between groups The statement "However, more CFS/ME patients who provided 7- day dietary feedback reported multivitamin supplement and/or vitamin D supplement consumption than HCs and hence CFS/ME patients showed higher ($p = 0.05$) vitamin D intake through use of supplements compared with HC individuals (4.9 ± 9.7 and $0.6 \pm 1.7 \mu\text{g/day}$, respectively), (Table 2)." Is not clear and needs rephrasing. ie. more CFS patients completed the diary compared to controls... but then CFS patients had higher VIT D intake... shouldn't this be controlled.. ie. you would exclude all those that didn't complete the food diary for that part of the analysis. These findings start to become a bit confusing and need clarification (1) what data you were able to collect. And then (2) how you analysed this. Attempting to summarise both in the same sentences etc. I think is what causes the confusion.

Again this would benefit in methodology if you clearly list what all your outcome measures are (ie. data you collected).

The authors agree that this section was not clear and so this has now been clarified (page 8).

Discussion:

Avoid referral to “sufferers” in reporting as this is considered discriminatory writing in scientific reporting. More appropriate is patients or individuals.

The authors are aware of the sensitivity surrounding CFS/ME and would not knowingly present a manuscript containing discriminatory writing. The term sufferers has been removed and altered accordingly.

I think the main point is that even though this initial study has shown any differences between cases and controls, vitamin D deficiency should still be considered at the individual level when it comes to managing patient. I think this point really needs to be drawn out. ie. once you really do clarify that aim and make it consistent. These points will ring clear.

The authors are most grateful for these points, but we remain content with our conclusions and wording. There was no evidence of increased fatigue in the individuals with lower vitamin D metabolites.

VERSION 2 – REVIEW

REVIEWER	Jos WM van der Meer Radboud UMC Nijmegen The Netherlands
REVIEW RETURNED	09-May-2017

GENERAL COMMENTS	No further comments
---------------------